

# Age-related effects on dynamic postural stability and prefrontal cortex activation during precision fitting tasks

Jiahao Pan[1] and Hui Tang[2]

[1] Biomedical Engineering Doctoral Program, Boise State University, Boise, ID, United States of America
[2] Department of Kinesiology and Health Education, The University of Texas at Austin, Austin, TX, United States of America

## ABSTRACT

**Background.** Dynamic postural control is impaired in older adults, as evidenced from worse dynamic postural stability compared to young adults during upright stance while concurrent goal-directed tasks. Prefrontal cortex (PFC) is considered to play an important role in goal-directed tasks. This study aimed to investigate the age effects on dynamic postural stability and PFC activation during precision fitting tasks.

**Methods.** Participant performed precision fitting tasks under four different conditions: large opening size with their arm's length (close-large), small opening size with their arm's length (close-small), large opening size with 1.3 times arm's length (far-large), and small opening size with 1.3 times arm's length (far-small). We analyzed the center of pressure-related outcomes representing dynamic postural stability and PFC activation at the six different subregions from healthy older adults ($n = 15$, $68.0 \pm 3.5$ years), and gender-matched middle-aged ($n = 15$, $48.73 \pm 3.06$ years) and young ($n = 15$, $19.47 \pm 0.64$ years) adults.

**Results.** The dynamic postural stability presented the young > middle-aged > older groups across the conditions. Specifically, the young group presented better dynamic postural stability than the older group in the close-large, far-large, and far-small conditions ($p < .05$), while showed better dynamic postural stability than the middle-aged group in the close-large condition ($p < .05$). Additionally, the older group had greater PFC activation at all PFC subregions than the young group ($p < .05$), while had greater activation at left dorsolateral and ventrolateral PFC than the middle-aged group ($p < .05$). The middle-aged group presented greater activation at left dorsomedial PFC than the young group ($p < .05$).

**Conclusion.** Heightened dorsomedial PFC activation in middle-aged adults compared to young adults may reflect a deficit in processing the visuomotor information during the precision fitting tasks. Degeneration of the ability in automatic coordination of dynamic postural control may begin to occur at midlife.

# INTRODUCTION

The ability to adopt and adjust posture while maintaining an upright stance and performing concurrent goal-directed tasks, such as reaching, grasping, and fitting, is crucial for daily

Corresponding author
Hui Tang, hui.tang@utexas.edu

activities (*Clark, Czaja & Weber, 1990*; *Tedeschi, 2023*). Aging, however, is associated with a noticeable decline in this dynamic postural control (*Bouisset & Do, 2008*). With age, individuals tend to shift from a flexible to a stiffer, less adaptive posture during such tasks, posing challenges in mitigating perturbations and sustaining dynamic postural stability (*Haddad et al., 2013*). Older adults exhibit worse dynamic postural stability than young adults, evidenced by greater center of pressure (CoP) displacement and sway area, along with reduced CoP trajectory smoothness during the goal-directed tasks with upright stance (*Huang & Brown, 2013*; *McNevin, Weir & Quinn, 2013*). This deficit in dynamic postural control may indicate more fall risks in older populations (*Haddad et al., 2013*). Age-related loss of postural automaticity may be associated with the deficits in dynamic postural control (*Clark, 2015*). Older adults may allocate additional attention-demanding executive control resources to compensate for deficits in dynamic postural control (*Clark, 2015*). Therefore, understanding the neural processing behind this age-related dynamic postural instability is vital for developing effective interventions to enhance dynamic postural control and minimize fall risks for the elderly, particularly in complex, multitasking scenarios.

The prefrontal cortex (PFC) plays an important role in processing the goal-directed tasks (*Funahashi & Andreau, 2013*; *Fuster & Bressler, 2015*; *Glover, Wall & Smith, 2012*; *Kaller et al., 2011*; *Mederos et al., 2021*; *Schöner, Bildheim & Zhang, 2024*; *Yamagata et al., 2012*). In general, increased in PFC activation would linked to greater degree of attention-demanding executive control adjustment during the complexed goal-directed tasks (*Mansouri, Tanaka & Buckley, 2009*). It is because automatic or previously learned behaviors may no longer achieve the given task (*Mansouri, Tanaka & Buckley, 2009*). Different PFC sub-regions, like the dorsolateral (engaged in retrieving behavioral-goal information) and ventrolateral (involved in encoding object features) areas, would process different information during the goal-directed tasks (*Yamagata et al., 2012*). Also, the dorsomedial region of PFC is key in dynamically reconfiguring visuomotor-related functional connectivity networks, integrating sensory input and motor planning for precise coordination during the goal-directed tasks (*Brovelli et al., 2017*). Therefore, monitoring the PFC activation in the different subregions would reflect age-related change in ability to process different information during the goal-directed tasks.

Postural automaticity reflects the ability of coordinating dynamic postural control with minimal use of attention-demanding executive control resources (*Anderson, 2018*; *Gaveau et al., 2014*; *Huang & Brown, 2015*; *Potocanac & Duysens, 2017*). Prior MRI study reported that performing simple reaching and grasping movements did not engage the PFC activation compared to planning without executing them. This finding indicated the automatic nature (*Glover, Wall & Smith, 2012*). In such condition, individuals' postural response could be rapidly and flexibly altered to adapt the concurrent arm movement (*Galgon, Shewokis & Tucker, 2010*; *Huang & Brown, 2015*; *Leonard et al., 2011*; *Lowrey, Nashed & Scott, 2017*). For instance, healthy young to middle-aged adults present better dynamic postural stability and higher hand accuracy in response to the goal-directed tasks after enough practice (*Galgon, Shewokis & Tucker, 2010*). However, the age-related deficits in generating appropriate postural response to achieve the optimized online movement trajectory was observed when performing the goal-directed tasks (*Goodman et al., 2018*;

*Goodman & Tremblay, 2021*; *Haaland, Harrington & Grice, 1993*; *Sarlegna, 2006*). As aging advances, older adults need longer planning durations, exhibit worse dynamic postural stability, and rely more on sensory feedback and cognitive processing to compensate the planning deficiencies and to correct the online movement trajectory of the goal-directed tasks (*Goodman & Tremblay, 2021*; *Huang & Brown, 2015*; *Sarlegna, 2006*). Although these studies highlight the significant role of brain in processing the goal-directed tasks, they have not directly examined the altered physiological brain function, particularly in the PFC (*Goodman & Tremblay, 2021*; *Huang & Brown, 2015*). Therefore, there is a research question whether older adults greater PFC activation and worse dynamic postural stability compared to young and middle-aged adults in response to the goal-directed tasks during upright standing.

Throughout the lifespan, the high-order cognitive function and postural control system have shown gradual degeneration, starting in the midlife and continuing thereafter (*Era et al., 2006*; *Peters, 2006*). For example, middle-aged adults showed reliance more on attention-demanding executive control resources assessed by greater PFC activation compared to young adults during cognitive memory tasks (*Klaassen et al., 2016*; *Kwon et al., 2016*). In terms of postural control, a slight but noticeable decline in balance is observed in adults aged 40 to 49 compared to those 30 to 39, with significant deterioration after 60 years old (*Era et al., 2006*). It is reasonable to suggest that the degeneration of automatic coordination of dynamic postural control in goal-directed tasks may also commence in midlife.

This study investigated the age-related effect on the PFC activation and dynamic postural stability during the precision fitting tasks. We hypothesized that (1) older adults would present a higher PFC activation and worse dynamic postural stability than the middle-aged and young adults during the precision fitting task; and (2) middle-aged adults would also show a higher PFC activation than the young adults, but no significant difference in dynamic postural stability during the precision fitting task.

## METHODS

### Participants

We recruited right-handed healthy young, middle-aged, and older adults in the current study. The inclusion criteria were as follows: (1) age 60 or older in the older group, age from 45 to 55 in the middle-aged group, age from 18 to 22 in the young group; (2) able to stand and walk at least 2 min without any assistance; (3) no injury or surgery at lower extremity in the past 6 months; (4) no neurological diseases, such as mild cognitive impairment, permanent memory loss, stroke, Parkinson's disease, and brain tumors; (5) Mini-Mental State Examination (MMSE) score $\geq$ 24 for all three groups, except for participants with 0 to 6 years of schooling (MMSE score $\geq$ 22) (*Cui et al., 2011*); and (6) no history of drug and alcohol abuse. The authors have permission to use MMSE from the copyright holders. This study was approved by the Ethics Committee of the Affiliated Hospital of Yangzhou University (2020-YKL12-23-02). Each participant signed the informed consent form before participation.

## Instrumentation setup

A custom-built instrument was utilized in this study, featuring a large whiteboard with two openings: a larger one measuring 130 × 130 mm and a smaller one measuring 100 × 100 mm. These openings were positioned in the upper middle section of the whiteboard, spaced 150 mm apart. Additionally, a fitting block measuring 90 × 90 mm, equipped with a cylindrical handle (20 mm in length and 10 mm in diameter), was placed on a custom-built base situated on a small table. Both the fitting board and the small table were designed with adjustable heights and positions. Finally, four pairs of optical sensors were employed, which are synchronized with Vicon Nexus system (Version 2.5, Vicon, Inc., Oxford, UK). Of these, two pairs were affixed to the top middle edge of each opening and on the whiteboard's backside, while the remaining pairs were attached to each side of the custom-built base.

The CoP data was collected using an embedded force plate (Kistler 9285BA, Kistler Corporation, Winterthur, Switzerland) at a sampling rate of 2,000 Hz. Data collection from the force plate was conducted using Vicon Nexus system (Version 2.5, Vicon, Inc., Oxford, UK). Changes of the cortical activation in the PFC were measured using an fNIRS device (Brite 24, Artinis medical systems, Einsteinweg, Netherlands) at a sampling rate of 50 Hz, utilizing two wavelengths of near-infrared light (760 and 850 nm). The setup included 10 sources and eight detectors, constituting 24 channels in total, positioned on the head's surface *via* a standard NIRS cap (10–10 international system) covering the PFC. The differential pathlength factor (DPF) was calculated was age-dependent: for participants younger than 55 years, DPF was determined using the formular: $4.99 + 0.067 \times Age^{0.814}$ (*Duncan et al., 1996*). For participants older than 55, the DPF was fixed set to 6.61 (*Claassen, Levine & Zhang, 2009*). To identify the subregions of PFC, five anatomical landmarks (nasion, inion, Cz, left and right preauricular points) were digitized using a Polhemus digitizer. Oxysoft was used for the collection prefrontal cortex activation. All devices were synchronized.

## Study protocol

This study was a cross-sectional design. Participants performed the precision fitting task in this study. The precision fitting task entails both the execution of typical goal-directed task and the need to maintain dynamic postural stability (*Pan et al., 2020*; *Pan et al., 2016*). Moreover, task constraints could be simply manipulated by decreasing the opening size (enhancing the fitting precision) and increasing the opening distance (enhancing the postural constraint) (*Coats et al., 2016*; *Huang & Brown, 2013*; *Potocanac & Duysens, 2017*; *Sarlegna, 2006*).

Participants wore uniform socks provided by the laboratory. Each participant stood on the center of the force plate with feet forming a 30° angle, heels being apart at 8% of the height, arms relax on each side, and align their middle line with the opening's center (*Pan et al., 2020*; *Pan et al., 2016*). Then, the height of opening was adjusted to the participant's shoulder height and aligned with the midline of body, and the distance of the board was adjusted to either their arm's length or 1.3 times arm's length, and the small table's height to the participant's waist height. Participants were instructed to fit the block into either

a large or small opening on the custom-made board under upright stance, positioned either an arm's length or at 1.3 times an arm's length from the board. Thus, four different conditions were performed by participants, including close-large, close-small, far-large, and far-small conditions. Participants began with the close condition, and the size condition was randomly selected. The order was counterbalanced across participants. Moreover, the purpose of performing close conditions first was to avoid participants starting with the most difficult far-small condition. It could minimize the learning effects that affects the outcomes in the close conditions. If participants either moved their feet or if the block touched the edge of the opening during the fitting task, the trial was categorized as "failed". For each condition, participants were required to complete five consecutive successful trails under the supervision of our experimental operator. Once a "failed" trial occurred, the participant was instructed to redo the five trials. Meanwhile, one experimental operator was required to stand in the back of the board, take the block, and put it back on the base as soon as possible. A 10-second baseline of quiet standing was recorded before the fitting task. Participants were given at least a 2 min break between conditions.

### Data analysis

The Vicon Nexus system was used to preprocess the CoP data, which was filtered with a fourth-order Butterworth low-pass filter with a cutoff frequency of 10 Hz (*Rocchi, Chiari & Horak, 2002*) and then exported in the format of CSV file for further analysis. Then, CoP data was divided into five trials based on the event signals from the optical sensors, where the onset of each trial was defined as the moment of rising the block and ending once the block completely passed through the opening. The standard deviation (SD) of CoP (CoP variability, $SD_{AP}$ & $SD_{ML}$), average CoP velocity ($V_{AP}$ & $V_{ML}$) at anterior-posterior (AP) and medial-lateral (ML) directions, and the 95% ellipse of sway area (SA) were calculated using the MATLAB (2021b, MathWorks, Natick, MA, USA). All these dependent variables were averaged over five trials under each condition. The higher value of these dependent variables represents worse dynamic postural stability.

To analyze the fNIRS data, the trail was defined as the first raising the block to the fifth passing the block through the opening under each condition based on the event signals from the optical sensors. The average duration of each condition was $19.74 \pm 2.35$ s, $24.88 \pm 3.51$ s, $22.45 \pm 3.11$ s and $30.47 \pm 4.03$ s in the young group, $19.77 \pm 2.10$ s, $28.39 \pm 3.16$ s, $24.24 \pm 3.91$ s and $33.15 \pm 6.35$ s in the middle-aged group, and $21.48 \pm 3.47$ s, $30.24 \pm 7.94$ s, $26.97 \pm 4.47$ s and $35.41 \pm 8.50$ s in the older group during the close-large, close-small, far-large, and far-small conditions, respectively. In the current study, only oxygenated hemoglobin ($HbO_2$) data were used for further analysis. It is because that a superior signal-to-noise ratio was observed for $HbO_2$ than deoxygenated hemoglobin (HHb) signals (*Strangman et al., 2002*) and $HbO_2$ and HHb were negatively associated (*Cui, Bray & Reiss, 2010*). The coefficients of variation (CV) for each channel of every participant were computed. Channels presenting a CV greater than 15% were excluded from subsequent data analyses, as they may include physical artifacts (*e.g.*, motion-induced instabilities of the coupling efficiency at the tissue-optical interfaces) and physiological artifacts (*e.g.*, blood-pressure-induced hemodynamics) (*Pinti et al., 2019*).

The preprocessing of the PFC activation used the Homer3 toolbox within MATLAB (The MathWorks, Inc. Natick, MA, USA), following distinct steps outlined in previous studies (*Koren, Parmet & Bar-Haim, 2019*; *Öztürk et al., 2021*): (1) converting raw data to optical density data; (2) removal of motion artifacts using the principal component analysis (tMotion = 1.0, tMask = 1.0, StdevThresh = 50 and AmpThresh = 0.5); (3) correction of motion artifacts using the spline interpolation ($p = 0.99$); (4) correction of motion artifacts using the wavelet based filter (iqr = 0.1); (5) corrected of physiological artifacts using the band-pass filter at a cutoff frequency of 0.01–0.14 Hz; (6) baseline-corrected by subtracting each trial individually from the 5 s of quiet standing; and (7) converting optical density data to concentrations. The $HbO_2$ concentration signals were exported to the TXT file to further calculate. According to the channels' positions, the average concentration of $HbO_2$ on six regions of interest (Fig. 1), including right dorsolateral PFC ($R\_PFC_{DL}$), left $PFC_{DL}$ ($L\_PFC_{DL}$), right dorsomedial PFC ($R\_PFC_{DM}$), left $PFC_{DM}$ ($L\_PFC_{DM}$), right ventrolateral PFC ($R\_PFC_{VL}$) and left $PFC_{VL}$ ($L\_PFC_{VL}$), were calculated.

## Statistical analysis

The two sets of dependent variables are dynamic postural stability, which includes $SD_{AP}$, $SD_{ML}$, $V_{AP}$, $V_{ML}$ and SA; and PFC activation, which includes $HbO_2$ concentration in the $R\text{-}PFC_{DL}$, $L\text{-}PFC_{DL}$, $R\text{-}PFC_{DM}$, $L\text{-}PFC_{DM}$, $R\text{-}PFC_{VL}$, and $L\text{-}PFC_{VL}$. The errors in fNIRS data are not independent across measurement channels (*Huppert, 2016*). Therefore, the Shapiro–Wilk's test was only used to assess the normality of each dependent variable of dynamic postural stability ($\alpha = 0.05$). Additionally, the multivariate normality and outliers of two sets of dependent variables were examined using Mahalanobis' distance.

Then, two two-way repeated measure MANOVAs (between-subject factor: group & within-subject factor: condition) were performed to investigate the effects of group and condition on dynamic postural stability and PFC activation, respectively. When a repeated measure MANOVA was significant, follow-up repeat measure ANOVAs and pairwise comparisons with Bonferroni correction were employed to detect significant main effects (group and condition) and interaction effect (group × condition) for each dependent variable. Partial Eta Squared ($\eta^2 p$) was calculated for overall effects and interactions for MANOVAs and ANONAs, where $\eta^2 p = .010$ is considered a small effect size, .060 medium effect size, and .14 or higher large effect size (*Cohen, 2013*). Cohen's d effect size was calculated to interpret the magnitude of specific *post hoc* comparisons, with $d = .20$ considered a small effect size, .50 medium effect size, and .80 or higher a large effect size (*Cohen, 2013*). The significant level was set at .05. All statistical analyses were performed using SPSS (28.0, IBM Inc., Armonk, NY, USA).

## RESULTS

The F-test power analysis was conducted using G*Power software (Version 3.1) with repeated measures MANOVA (within-between interaction) to determine the minimal sample size. The effect size for comparing cortical activation between single-task and dual-task paradigms in older adults ranges from moderate to strong based on prior works (*Pan & Zhang, 2024*; *Salzman et al., 2021*). Therefore, we set the effect size at .65.

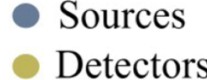

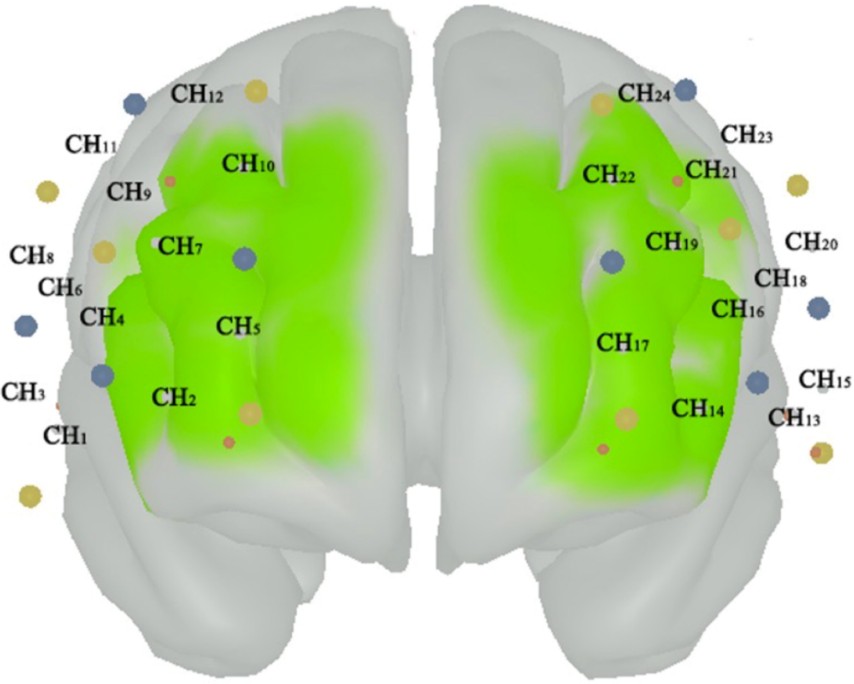

**Figure 1** **The position of composed channels by the sources (blue color) and detectors (funky yellow color) in prefrontal cortex.** The right dorsolateral prefrontal cortex (PFC) includes the $CH_3$, $CH_6$, $CH_8$, $CH_9$ and $CH_{11}$; the left dorsolateral PFC includes the $CH_{15}$, $CH_{18}$, $CH_{20}$, $CH_{21}$ and $CH_{23}$; the right dorsomedial PFC includes the $CH_2$, $CH_5$, $CH_7$, $CH_{10}$ and $CH_{12}$; the left dorsomedial PFC includes the $CH_{14}$, $CH_{17}$, $CH_{19}$, $CH_{22}$ and $CH_{24}$; the right ventrolateral PFC includes the $CH_1$ and $CH_4$; and the left ventrolateral PFC includes the $CH_{13}$ and $CH_{16}$.

**Table 1** **Mean ± standard deviation values of demographic information and Mini-Mental State Examination (MMSE) score.**

|  | Older group | Middle-aged group | Young group |
|---|---|---|---|
| Age (years) | 68.07 ± 3.58 | 48.73 ± 3.06 | 19.47 ± 0.64 |
| Height (cm) | 159.93 ± 8.43 | 157.80 ± 8.09 | 168.73 ± 6.88 |
| Weight (kg) | 60.71 ± 9.20 | 61.99 ± 9.99 | 64.77 ± 14.26 |
| MMSE | 25.33 ± 1.54 | 25.40 ± 1.40 | 28.33 ± 1.50 |

Additionally, the analysis parameters included an alpha level of .05, a statistical power of .80, three groups, and six measurements. The results suggested a minimum sample size of 25 participants, with at least nine participants per group. In the current study, each group included seven males and eight females. Mean ± standard deviation values of demographic information and MMSE score were presented in Table 1.

**Table 2  Mean ± standard deviation values of dynamic posture stability when performing the precision fitting task among the older, middle-aged, and young groups under different conditions.**

| Variables | Older group | Middle-aged group | Young group |
|---|---|---|---|
| | Close-large condition | | |
| $SD_{AP}$ G, C & I | 34.48 ± 10.94 | 32.60 ± 8.83 | 21.89 ± 8.07 |
| $SD_{ML}$ G & C | 13.95 ± 8.62 | 8.84 ± 3.68 | 6.05 ± 2.96 |
| $V_{AP}$ (cm/s) G, C & I | 7.45 ± 2.71 | 7.41 ± 2.47 | 5.27 ± 1.70 |
| $V_{ML}$ (cm/s) C & I | 3.83 ± 1.32 | 3.18 ± 1.28 | 2.66 ± 1.08 |
| SA ($cm^2$) G & C | 39.73 ± 22.70 | 30.56 ± 19.99 | 17.10 ± 11.15 |
| | Close-small condition | | |
| $SD_{AP}$ G & C | 24.12 ± 5.96 | 20.79 ± 4.51 | 20.28 ± 8.75 |
| $SD_{ML}$ G & C | 10.01 ± 6.52 | 5.98 ± 3.72 | 4.91 ± 2.25 |
| $V_{AP}$ (cm/s) G & C | 3.84 ± 1.01 | 3.45 ± 0.52 | 3.58 ± 1.12 |
| $V_{ML}$ (cm/s) C | 2.35 ± 1.35 | 1.77 ± 0.46 | 2.03 ± 0.90 |
| SA ($cm^2$) G & C | 23.34 ± 10.59 | 14.75 ± 7.47 | 13.74 ± 8.35 |
| | Far-large condition | | |
| $SD_{AP}$ G, C & I | 41.95 ± 9.99 | 35.81 ± 9.56 | 29.19 ± 11.04 |
| $SD_{ML}$ G & C | 14.55 ± 5.51 | 12.23 ± 3.95 | 10.57 ± 3.97 |
| $V_{AP}$ (cm/s) G & C | 8.99 ± 2.24 | 7.65 ± 2.45 | 7.71 ± 2.19 |
| $V_{ML}$ (cm/s) C | 4.53 ± 1.57 | 4.15 ± 1.28 | 4.78 ± 1.71 |
| SA ($cm^2$) G & C | 69.53 ± 30.44 | 47.61 ± 20.52 | 39.59 ± 23.27 |
| | Far-small condition | | |
| $SD_{AP}$ G, C & I | 39.15 ± 8.35 | 33.65 ± 12.14 | 24.59 ± 12.45 |
| $SD_{ML}$ G & C | 18.17 ± 13.65 | 10.29 ± 4.11 | 7.61 ± 3.58 |
| $V_{AP}$ (cm/s) G, C & I | 6.17 ± 1.22 | 5.16 ± 1.41 | 4.31 ± 1.66 |
| $V_{ML}$ (cm/s) C & I | 3.83 ± 1.47 | 2.88 ± 0.81 | 2.70 ± 1.24 |
| SA ($cm^2$) G & C | 69.25 ± 41.47 | 41.31 ± 26.51 | 26.36 ± 22.99 |

**Notes.**

$SD_{AP}$ means CoP variability in the AP direction; $SD_{ML}$ means CoP variability in the ML direction; $V_{AP}$ means average CoP velocity in the AP direction; $V_{ML}$ means average CoP velocity in the ML direction; and SA means sway area.

G Indicates a significant group difference. C Indicated a significant condition difference. I Indicated a significant interaction difference.

## Dynamic postural stability

Some dependent variables of dynamic postural stability presented non-normal distribution based on the Shapiro–Wilk's test. These non-normally distributed dependent variables were log10 transformed before statistical analysis. Additionally, our dependent variables of the dynamic postural stability were multivariate normally distributed (MD < 20.52). Table 2 showed the Mean ± standard deviation values of dynamic postural stability among three groups under different conditions.

There were significant group (Wilk's lambda = .455, $F(10, 76) = 3.672$, $p < .001$, $\eta^2 p$ = .326) and condition (Wilk's lambda = .048, $F(15, 28) = 36.892$, $p < .001$, $\eta^2 p = .952$) effects, and group × condition interaction (Wilk's lambda = .217, $F(30, 56) = 2.139$, $p = .007$, $\eta^2 p = .534$) effect on the association of dependent variables. Based on follow-up ANOVA with repeated measure tests, the significant effects of group, condition, and group × condition interaction were observed in the $SD_{AP}$ (group effect: $F(2, 42) = 8.926$, $p = .001$,

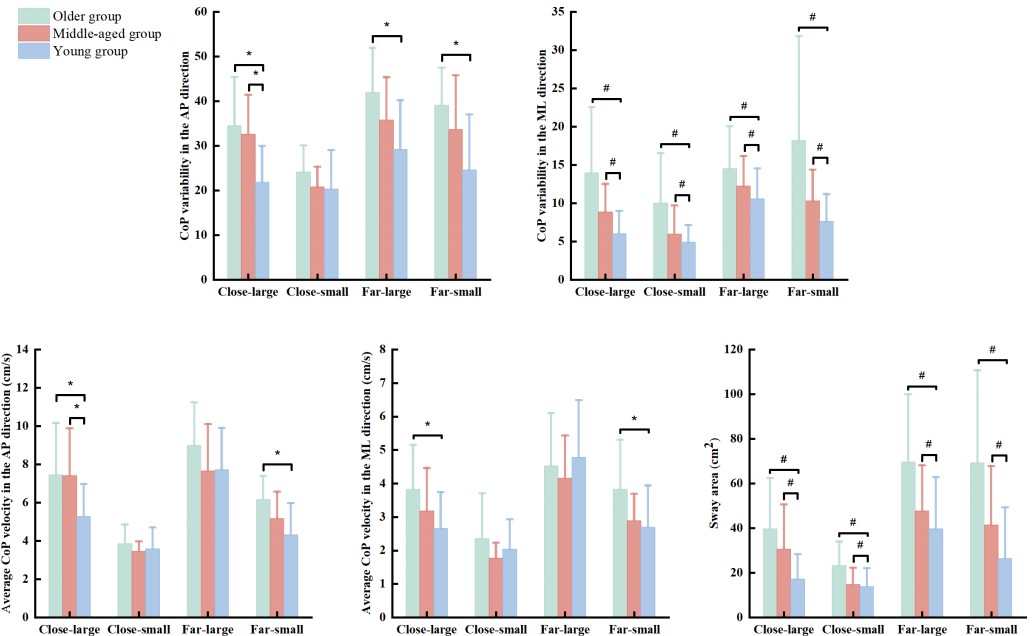

**Figure 2** **The significant group and interaction effects in the dynamic posture stability among young, middle-aged, and older group during precision fitting task.** CoP means center of pressure; AP means anterior-posterior; and ML means medial-lateral. * Indicates a significant interaction difference among the groups under different conditions. # Indicates a significant difference compared to the older group regardless of the conditions.

$\eta^2 p = .298$; condition effect: $F(3, 126) = 42.311$, $p < .001$, $\eta^2 p = .502$; & interaction effect: $F(6, 126) = 2.819$, $p = .020$, $\eta^2 p = .118$) and $V_{AP}$ (group effect: $F(2, 42) = 4.129$, $p = .023$, $\eta^2 p = .164$; condition effect: $F(3, 126) = 136.575$, $p < .001$, $\eta^2 p = .765$; & interaction effect: $F(6, 126) = 3.960$, $p = .002$, $\eta^2 p = .159$) (Table 2). Additionally, there was significant condition effect ($F(3, 126) = 91.357$, $p < .001$, $\eta^2 p = .952$) and group × condition ($F(6, 126) = 2.680$, $p = .021$, $\eta^2 p = .113$) interaction in the $V_{ML}$ (Table 2). *Post hoc* analysis indicated that the older group presented greater $SD_{AP}$ than the young group in the close-large ($p = .002$ & Cohen's $d = 1.31$), far-large ($p = .002$ & Cohen's $d = 1.21$), and far-small ($p < .001$ & Cohen's $d = 1.37$) conditions (Fig. 2). Also, the older group showed greater $V_{AP}$ and $V_{ML}$ than the young group in the close-large ($V_{AP}$: $p = .029$ & Cohen's $d = 1.73$ & $V_{ML}$: $p = .034$ & Cohen's $d = 0.97$) and far-small ($V_{AP}$: $p = .001$ & Cohen's $d = 1.27$ & $V_{ML}$: $p = .021$ & Cohen's $d = 0.83$) conditions. Additionally, the middle-aged group showed grater $SD_{AP}$ ($p = .005$ & Cohen's $d = .77$) and $V_{AP}$ ($p = .029$ & Cohen's $d = .77$) compared to the young group in the close-large condition (Fig. 2).

The ANOVA with repeated measure tests further observed group and condition effects in the $SD_{ML}$ (group effect: $F(2, 42) = 10.919$, $p < .001$, $\eta^2 p = .342$; & condition effect: $F(3, 126) = 27.039$, $p < .001$, $\eta^2 p = .392$) and SA (group effect: $F(2, 42) = 13.014$, $p < .001$, $\eta^2 p = .383$; & condition effect: $F(3, 126) = 58.553$, $p < .001$, $\eta^2 p = .582$). *Post hoc* test reported that the older group showed greater $SD_{ML}$ (young group: $p < 0.001$ & Cohen's $d = .96$ & middle-age group: $p = 0.028$ & Cohen's $d = .66$) and SA (young group: $p < 0.001$

**Table 3** Mean ± standard deviation values of HbO 2 (μm/ml) in the prefrontal cortex when performing the precision fitting task among the older, middle-aged, and young groups under different conditions.

| Variables | Older group | Middle-aged group | Young group |
|---|---|---|---|
| | | Close-large condition | |
| R_PFC$_{DL}$$^{G\&C}$ | 0.28 ± 0.23 | 0.15 ± 0.22 | 0.042 ± 0.18 |
| L_PFC$_{DL}$$^{G}$ | 0.28 ± 0.20 | 0.082 ± 0.15 | 0.036 ± 0.24 |
| R_PFC$_{DM}$$^{G\&C}$ | 0.24 ± 0.23 | 0.10 ± 0.22 | −0.011 ± 0.19 |
| L_PFC$_{DM}$$^{G\&C}$ | 0.22 ± 0.18 | 0.10 ± 0.18 | 0.025 ± 0.26 |
| R_PFC$_{VL}$$^{G\&C}$ | 0.35 ± 0.28 | 0.29 ± 0.34 | 0.048 ± 0.33 |
| L_PFC$_{VL}$$^{G\&C}$ | 0.31 ± 0.29 | 0.14 ± 0.19 | 0.023 ± 0.32 |
| | | Close-small condition | |
| R_PFC$_{DL}$$^{G\&C}$ | 0.51 ± 0.33 | 0.33 ± 0.61 | 0.10 ± 0.19 |
| L_PFC$_{DL}$$^{G}$ | 0.45 ± 0.32 | 0.27 ± 0.52 | 0.048 ± 0.17 |
| R_PFC$_{DM}$$^{G\&C}$ | 0.42 ± 0.31 | 0.25 ± 0.60 | 0.15 ± 0.19 |
| L_PFC$_{DM}$$^{G\&C}$ | 0.42 ± 0.31 | 0.32 ± 0.55 | −0.056 ± 0.30 |
| R_PFC$_{VL}$$^{G\&C}$ | 0.65 ± 0.43 | 0.47 ± 0.95 | 0.19 ± 0.31 |
| L_PFC$_{VL}$$^{G\&C}$ | 0.60 ± 0.43 | 0.37 ± 0.56 | 0.068 ± 0.23 |
| | | Far-large condition | |
| R_PFC$_{DL}$$^{G\&C}$ | 0.53 ± 0.57 | 0.23 ± 0.32 | 0.14 ± 0.39 |
| L_PFC$_{DL}$$^{G}$ | 0.53 ± 0.53 | 0.12 ± 0.36 | 0.061 ± 0.27 |
| R_PFC$_{DM}$$^{G\&C}$ | 0.51 ± 0.57 | 0.21 ± 0.36 | 0.12 ± 0.34 |
| L_PFC$_{DM}$$^{G\&C}$ | 0.48 ± 0.52 | 0.22 ± 0.39 | −0.066 ± 0.34 |
| R_PFC$_{VL}$$^{G\&C}$ | 0.73 ± 0.64 | 0.38 ± 0.55 | 0.18 ± 0.50 |
| L_PFC$_{VL}$$^{G\&C}$ | 0.61 ± 0.59 | 0.27 ± 0.43 | −0.071 ± 0.64 |
| | | Far-small condition | |
| R_PFC$_{DL}$$^{G\&C}$ | 0.54 ± 0.33 | 0.49 ± 0.30 | 0.35 ± 0.42 |
| L_PFC$_{DL}$$^{G}$ | 0.47 ± 0.41 | 0.25 ± 0.29 | 0.17 ± 0.31 |
| R_PFC$_{DM}$$^{G\&C}$ | 0.54 ± 0.29 | 0.48 ± 0.33 | 0.27 ± 0.46 |
| L_PFC$_{DM}$$^{G\&C}$ | 0.68 ± 0.41 | 0.42 ± 0.39 | 0.14 ± 0.43 |
| R_PFC$_{VL}$$^{G\&C}$ | 0.84 ± 0.48 | 0.75 ± 0.57 | 0.57 ± 0.64 |
| L_PFC$_{VL}$$^{G\&C}$ | 0.96 ± 0.53 | 0.62 ± 0.58 | 0.37 ± 0.60 |

**Notes.**

R means right; L means left; PFC$_{DL}$ means dorsolateral prefrontal cortex; PFC$_{DM}$ means dorsomedial prefrontal cortex; PFC$_{VL}$ means ventrolateral prefrontal cortex.

[G] Indicates a significant group difference. [C] Indicated a significant condition difference. [I] Indicated a significant interaction difference.

& Cohen's $d = .94$ & middle-age group: $p = 0.030$ & Cohen's $d = .58$) than the young and middle-aged groups across the conditions (Fig. 2). We did not report the condition effect since our interests were the effects of group and interaction between group and condition.

## Prefrontal cortex activation

Our dependent variables of the PFC activation were multivariate normally distributed within each group of the independent variables (MD < 22.46). Table 3 showed the mean ± standard deviation values of HbO$_2$ concentration in the PFC among three groups under different conditions.

There were significant group (Wilk's lambda $= .467$, $F(12, 74) = 3.455$, $p = .003$, $\eta^2 p = .317$) and condition (Wilk's lambda $= .295$, $F(18, 25) = 3.327$, $p = .003$, $\eta^2 p = .75$) effects on the association of dependent variables. No significant group $\times$ condition interaction effect (Wilk's lambda $= .253$, $F(36, 50) = 1.371$, $p = .150$, $\eta^2 p = .497$) was observed in the MANOVA model. Follow-up ANOVA with repeated measure tests reported group effect in the R_PFC$_{DL}$ ($F(2, 42) = 7.693$, $p = .001$, $\eta^2 p = .268$), L_PFC$_{DL}$ ($F(2, 42) = 11.076$, $p < .001$, $\eta^2 p = .345$), R_PFC$_{DM}$ ($F(2, 42) = 7.701$, $p = .001$, $\eta^2 p = .268$), L_PFC$_{DM}$ ($F(2, 42) = 15.637$, $p < .001$, $\eta^2 p = .427$), R_PFC$_{VL}$ ($F(2, 42) = 6.920$, $p = .003$, $\eta^2 p = .248$), and L_PFC$_{VL}$ ($F(2, 42) = 12.325$, $p < .001$, $\eta^2 p = .370$) and condition effect in the R_PFC$_{DL}$ ($F(3, 126) = 5.933$, $p = .002$, $\eta^2 p = .124$), R_PFC$_{DM}$ ($F(3, 126) = 6.390$, $p < .001$, $\eta^2 p = .132$), L_PFC$_{DM}$ ($F(3, 126) = 5.675$, $p = .001$, $\eta^2 p = .119$), R_PFC$_{VL}$ ($F(3, 126) = 6.935$, $p = .001$, $\eta^2 p = .142$), and L_PFC$_{VL}$ ($F(3, 126) = 10.674$, $p < .001$, $\eta^2 p = .203$) (Table 3). *Post hoc* test indicated that the older group presented greater HbO$_2$ activation in the R_PFC$_{DL}$ ($p = .001$ & Cohen's $d = .85$), L_PFC$_{DL}$ ($p < .001$ & Cohen's $d = 1.08$), R_PFC$_{DM}$ ($p = .001$ & Cohen's $d = .84$), L_PFC$_{DM}$ ($p < .001$ & Cohen's $d = 1.18$), R_PFC$_{VL}$ ($p = .002$ & Cohen's $d = .79$), and L_PFC$_{VL}$ ($p < .001$ & Cohen's $d = 1.04$) than the young group, while older group also showed greater HbO$_2$ concentration in the L_PFC$_{DL}$ ($p = .007$ & Cohen's $d = .38$) and L_PFC$_{VL}$ ($p = .041$ & Cohen's $d = .54$) than the middled-aged group across the conditions (Fig. 3). Also, the middle-aged group presented greater HbO$_2$ concentration in the L_PFC$_{DM}$ ($p = .008$ & Cohen's $d = .36$) than the young group (Fig. 3). We did not report the condition effect since our interests were the group and interaction effects among group and between group and condition.

## DISCUSSION

The current study aimed to explore the effects of precision fitting task on the dynamic posture stability and PFC activation at six different subregions among the young, middle-aged, and old groups. Our results indicated that (1) the older group presented worse dynamic postural stability compared to the young group in all of the conditions, except for the close-small condition; (2) the middle-age group only showed worse dynamic postural stability compared to the young group in the close-large condition; (3) regardless the conditions, the middle-aged group exhibited better dynamic postural stability compared to the older group; (4) the older group presented greater HbO$_2$ concentration in all PFC's subregions compared to the young group; and (5) the middle-age group showed lower HbO$_2$ concentration in the L_PFC$_{DL}$ and L_PFC$_{VL}$ compared to the older group, but they had greater HbO$_2$ concentration in the L_PFC$_{DM}$ compared to the young group. Our observations are consistent with our hypothesis.

The young group presented better dynamic postural stability compared to the middle-aged and older groups in the close-large condition. The observation is consistent with prior works (*Huang & Brown, 2013*; *McNevin, Weir & Quinn, 2013*; *Walz et al., 2023*). For instance, one study indicated that the gait speed in the timed up & go task with goal-directed arm-movement task presented faster in the young group compared to the

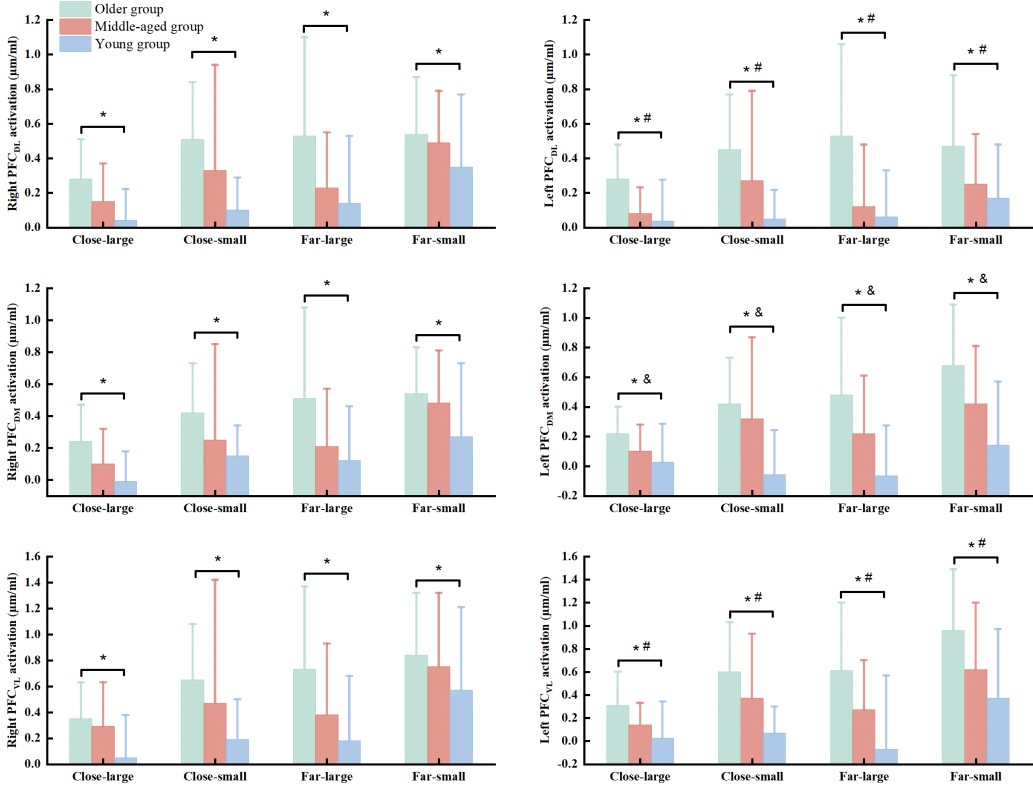

**Figure 3** **The significant group effect in the cortical activation among young, middled-aged, and older group during precision fitting task.** $PFC_{DL}$ means dorsolateral prefrontal cortex; $PFC_{DM}$ means dorsomedial prefrontal cortex; $PFC_{VL}$ means ventrolateral prefrontal cortex. [*] Indicates a significant difference between the older and young groups regardless of the conditions. [#] Indicates a significant difference between the older and middle-aged groups regardless of the conditions. [&] Indicates a significant difference between the middle-aged and young groups regardless of the conditions.

middle-aged and older groups (*Walz et al., 2023*). The possible explanation is a decline in controlling whole-body postural fluctuations with aging when the upright stance is perturbated by a precision goal-directed task (*Haddad et al., 2013*). Compared to close-large condition, the fitting task is becoming more difficult when decreasing the opening size (enhancing the fitting precision) and increasing the opening distance (enhancing the postural constraint) (*Coats et al., 2016*; *Huang & Brown, 2013*; *Potocanac & Duysens, 2017*; *Sarlegna, 2006*). To successfully complete these conditions, participants need to control body configurations to the limits of upright stability and to gear upper body actions to locate the object simultaneously (*Haddad et al., 2013*; *Ornkloo & Von Hofsten, 2007*). Increased task difficulty leads to an increase in completion time, less smooth of arm trajectory, worse endpoint accuracy, and long-latency response to arm movement (*Coats et al., 2016*; *Huang & Brown, 2013*; *Potocanac & Duysens, 2017*; *Sarlegna, 2006*). Interestingly, the current study did not observe the significant difference in dynamic postural stability between the young and middle-aged groups in the close-small, far-large, and far-small conditions. The middle-aged group presented better dynamic postural stability than older

adults across the conditions. An explanation is that the study protocol is fixed order from close to far conditions. Middle-aged adults could have the ability to learn fitting skills from close conditions, because previous study indicated that motor learning was significantly slower in adults over 62 years when learning novel visuomotor task (*Smith et al., 2005*). For the older adults, they presented worse dynamic postural stability than young adults in the far-large and far-small conditions. Previous work indicated that far conditions demand an increased degree of coordination between posture and manual motor task compared to close conditions (*Rossi, Mitnitski & Feldman, 2002*). Sensory systems are required to additionally process to update the position of trunk, hand and opening to optimize movement accuracy at far conditions (*Cheng et al., 2012*; *Goodman et al., 2018*). Thus, it might be the deficits in the capacity of using the existed visuospatial information or online visual control to guide the fitting task in older adults that induced the increased dynamic postural instability (*Cheng et al., 2012*; *Grabowski & Mason, 2014*). Another possible explanation is that older adults show decreasing the complexities of multi-joint movement *via* the strategy of freezing the trunk and arm, but is not allowed to suppress dynamic postural fluctuations and to speed up fitting movement among older adults (*Fuster & Bressler, 2015*; *Gaveau et al., 2014*; *Haddad et al., 2012*).

The current study also observed that older group had greater $HbO_2$ activation in the $PFC_{DL}$, $PFC_{DM}$, and $PFC_{VL}$ than young adults across conditions. Prior works have demonstrated that heightened $HbO_2$ activation in the PFC subregions ($PFC_{DL}$, $PFC_{DM}$, $PFC_{VL}$) in response to the goal-directed behaviors may be associated with processing visuospatial, visuomotor, and visual object information, respectively (*Yamagata et al., 2012*; *Brovelli et al., 2017*). The precision fitting task requires the sensory feedback from hand and target position and adapting postural configurations to optimize movement accuracy due to increasing the terminal accuracy (*Lowrey, Nashed & Scott, 2017*; *Sarlegna, 2006*; *Zhou et al., 2011*). Older adults have shown the impaired ability of processing the visuospatial and visuomotor information in goal-directed tasks (*Cheng et al., 2012*; *Grabowski & Mason, 2014*). Accordingly, we speculate that additional attention-demanding executive control resources are used to monitor the interaction among environment, arm and trunk movement, target, and upright stance in older adults. In addition, the middle-aged group presented greater $HbO_2$ activation in the $PFC_{DM}$ than young group across the conditions. Considering the function of increased $PFC_{DM}$, middle-aged adults may require substantial effort in integrating sensory input and motor planning for precise coordination in goal-directed tasks compared to young adults (*Brovelli et al., 2017*). These observations may suggest that middle-aged adults initially degenerate the ability in processing the visuomotor information, rather than visuospatial and visual object information (*Yamagata et al., 2012*; *Brovelli et al., 2017*). This suggestion is further supported by other results in the current study, which indicates smaller $HbO_2$ activation in the $PFC_{DL}$ and $PFC_{VL}$ in the middle-aged group compared to older group.

Taken the dynamic postural stability and $HbO_2$ activation in the PFC regions together, this study provide evidence indicating that loss of automaticity in coordination task-dependent postural control may emerges earlier in adulthood at midlife (*Potocanac & Duysens, 2017*; *Sarlegna, 2006*). It is because we observed greater $HbO_2$ activation in the

$PFC_{DM}$ and worse dynamic posture stability in middle-aged adults compared to young adults. In some contexts, heightened PFC activation in middle-aged adults is utilized to preserve dynamic postural stability compared to young adults in the close-small, close-far, and far-small conditions. Hence, middle-aged adults can be explained by the "compensation" hypothesis, which presents slight decline in their brain function and cognition (*Fettrow et al., 2021*; *Levin et al., 2014*). The older group presented greater $HbO_2$ activation in all the PFC subregions and worse dynamic postural stability compared to middle-aged and young groups. These observations might be supported by the "de-differentiation" hypothesis (*Levin et al., 2014*). Meanwhile, increased $HbO_2$ activation in the PFC regions fails to improve dynamic postural stability in older adults due to less specificity of PFC functions (*Fettrow et al., 2021*). These observations may imply the importance of neural processing at the highest levels of the control hierarchy in coordination dynamic postural control in goal-directed tasks. This is clinically important because it links the potential mechanism in loss of automatic coordination of dynamic postural control with aging and might be a strong predictor of risk of falls. Moreover, the degeneration of postural automaticity in middle-aged adults should not be overlooked. Future studies can focus on identifying rehabilitation protocols that boost the ability in mediating task planning and execution for cognitive and motor functions in both middle-aged and older groups.

This is the first study to simultaneously investigate PFC activation and dynamic postural stability in three different age groups (young, middle-aged, and older adults) when performing precision fitting task. However, the limitations of this study should not be overlooked. First, this study was not a randomized controlled trial. It may have introduced selection bias and affected our results. Second, the sample size is small, which limits the generalizability of the observations. Third, although fNIRS has good temporal specificity, it can only record cortical activation and restricting the region of interest to the PFC in this study. This prevents us from analyzing deeper regions such as subcortical structures as well as other higher-order cognitive regions such as the motor cortex.

## CONCLUSION

In the current study, the dynamic postural stability presented young group > middle-aged group > older group, which suggested that individuals reaching to middle-age is associated with an impaired ability in suppressing dynamic postural fluctuations during the precision fitting task. Additionally, middle-aged adults presented higher $HbO_2$ activation in the $PFC_{DM}$ than young adults, as well as showed lower $HbO_2$ activation in the $PFC_{DL}$ and $PFC_{VL}$ than older adults across the conditions. This observation may further suggest that individuals reaching middle-age are associated with an impaired ability in processing the visuomotor information during the goal-directed tasks. These observations are clinically important because they suggested that rehabilitation interventions improving the visuomotor-related function could improve the dynamic postural control and minimize the risk of falls.

### Funding
The authors received no funding for this work.

### Competing Interests
The authors declare there are no competing interests.

### Author Contributions
- Jiahao Pan conceived and designed the experiments, performed the experiments, analyzed the data, prepared figures and/or tables, authored or reviewed drafts of the article, and approved the final draft.
- Hui Tang conceived and designed the experiments, analyzed the data, prepared figures and/or tables, authored or reviewed drafts of the article, and approved the final draft.

### Human Ethics
The following information was supplied relating to ethical approvals (*i.e.*, approving body and any reference numbers):

This study was approved by the Ethics Committee of the Affiliated Hospital of Yangzhou University (2020-YKL12-23-02).

### Data Availability
The raw measurements are available in the Supplementary File.

### Supplemental Information
Supplemental information for this article can be found online at http://dx.doi.org/10.7717/peerj.18548#supplemental-information.

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
