# Peer review of "Age-related effects on dynamic postural stability and prefrontal cortex activation during precision fitting tasks"

_PeerJ, doi:10.7717/peerj.18548_

## Round 0.1 · original submission · Major Revisions

Both reviewers raised important comments regarding the methodological approach as well as the discussion and limitations of the study which need more elaboration.

Reviewer 1 ·

Basic reporting

Clear and Unambiguous, Professional English Used Throughout

Comment: The article generally maintains a professional tone and is written in clear English. However, there are some sections where the language could be refined for better clarity and precision. Specific terms should be consistently used throughout the paper to avoid confusion.

Literature References, Sufficient Field Background/Context Provided

Comment: The article provides an adequate background and cites relevant literature, but it could benefit from a more comprehensive review of recent studies(10.33607/rmske.v2i29.1428). The connection between the current research and existing literature could be strengthened to better highlight the study’s contribution.

3. Professional Article Structure, Figures, Tables, Raw Data Shared

Comment: The article follows a standard structure with clear sections, but the presentation of figures and tables could be improved for clarity.

4. Self-Contained with Relevant Results to Hypotheses

Comment: The article is mostly self-contained and presents results relevant to the stated hypotheses. However, some results could be discussed in more detail to better connect them to the hypotheses. The discussion should more thoroughly explore the implications of the findings in the context of the hypotheses tested.

Experimental design

Original Primary Research within Aims and Scope of the Journal

Comment: The research aligns well with the journal's aims and scope, focusing on age-related changes in postural stability and prefrontal cortex activation. The topic is relevant and contributes to the field of biomechanics and neuroscience.

Research Question Well Defined, Relevant & Meaningful

Comment: The research question is clearly defined and addresses a significant issue related to aging and motor control. The study successfully identifies a knowledge gap concerning the onset of cognitive decline in postural control and how this gap is addressed by the research.

Rigorous Investigation Performed to a High Technical & Ethical Standard

Comment: The study appears to have been conducted rigorously, with appropriate technical methods and adherence to ethical standards. However, some aspects, such as the fixed order of tasks, could potentially introduce biases that need further discussion.

Methods Described with Sufficient Detail & Information to Replicate

Comment: The methods are generally well-detailed, allowing for potential replication of the study. However, additional clarity on certain procedures, such as the justification for specific task conditions and statistical methods, would further enhance replicability.

Validity of the findings

Impact and Novelty Not Assessed. Meaningful Replication Encouraged

Comment: The study does not focus on assessing impact or novelty, which is appropriate given the journal's criteria. The research is valuable as it addresses a significant issue related to aging, with potential implications for replication studies. The rationale for the study is clearly stated, contributing meaningful insights to the literature on postural control and cognitive aging.

All Underlying Data Provided; Robust, Statistically Sound, & Controlled

Comment: The study provides the necessary data, and the analyses appear statistically sound and well-controlled. The data is robust, with appropriate measures taken to ensure reliability. However, the presentation of the data could be enhanced by providing more detailed explanations in the supplementary materials to facilitate further analysis by other researchers.

Conclusions Well Stated, Linked to Original Research Question & Limited to Supporting Results

Comment: The conclusions are clearly linked to the original research question and are appropriately supported by the results. The paper effectively limits its conclusions to the findings presented, avoiding overgeneralization. However, a more in-depth discussion of the broader implications of these conclusions could add value.

Additional comments

Clarity and Consistency in Language

Improvement Needed: Some sections of the manuscript would benefit from clearer and more consistent language. Specific terms should be used consistently throughout the paper to avoid confusion. Additionally, certain sentences could be rephrased for better clarity and readability.

Comprehensive Literature Review

Improvement Needed: The literature review, while adequate, could be expanded to include more recent studies. This would strengthen the background context and demonstrate a more thorough engagement with the current state of research in the field.

Justification of Methodological Choices

Improvement Needed: The rationale behind specific methodological choices, such as the selection of task conditions and the order in which they were presented, needs to be more explicitly discussed. This will help justify the experimental design and address potential biases.

Discussion Depth

Improvement Needed: The discussion section should be more comprehensive, exploring alternative explanations for the findings and considering the broader implications of the results. This would provide a deeper understanding of the study's significance and its contribution to the field.

Detailed Data Presentation

Improvement Needed: While the data is robust and statistically sound, its presentation could be improved. More detailed explanations of the data, especially in the supplementary materials, would make the findings more accessible and useful for other researchers.

Expanding on Limitations

Improvement Needed: The manuscript should include a more detailed discussion of its limitations. This could include the potential effects of the fixed task order, the relatively small sample size, and any other methodological constraints. Addressing these openly would strengthen the study's credibility.

Broader Implications of Conclusions

Improvement Needed: The conclusions are well stated but could be enhanced by discussing the broader implications of the findings more explicitly. This would help situate the study within the larger context of research on aging and motor control.

·

Basic reporting

- The article uses generally clear and professional language. However, some minor grammatical issues and awkward phrasing can be found throughout, which may hinder readability for an international audience. For example, phrases like "indictor of dynamic postural stability" and "heightened attention processing is used" could be revised for clarity. Additionally, some sentences are overly long and complex, which could be broken down to improve comprehension.

- The introduction provides a comprehensive background on the relationship between dynamic postural stability, prefrontal cortex activation, and aging. The literature is well referenced, covering relevant studies on neural mechanisms, age-related changes, and task-dependent postural control. However, the paper could benefit from further contextualization of the study’s contributions to existing knowledge gaps. While the background is solid, the rationale for the study could be emphasized more clearly. Specifically, a more explicit discussion of how this study advances or fills gaps in the current body of literature would strengthen the introduction.

- The article follows a logical structure with clearly defined sections. The use of figures and tables is appropriate for presenting data, but the quality and clarity of these visual aids could be improved. The raw data appears to be sufficiently provided in line with the journal's policies, although this should be verified upon reviewing supplementary materials.

- The article presents its findings in a self-contained manner, directly addressing the research hypotheses. However, the interpretation of results could be more tightly linked to the study's objectives, particularly in the discussion section. Some conclusions seem speculative without strong backing from the results presented.

Experimental design

- The study meets ethical standards and follows rigorous technical procedures, as evidenced by the use of well-established equipment and methods. However, an important omission is the declaration of the study type and design, which is critical for evaluating the rigor of the methodology. It is suggested to explicitly state the type of study and the design (in the methods section). This will allow the reader to clearly understand the structure of the research and improve its scientific rigor.

Validity of the findings

- Clearly state the potential impact and novelty of the findings in the discussion. Emphasize how this research can contribute to advancing the understanding of postural control and prefrontal cortex activation in different age groups, and suggest specific areas where future research can replicate or extend these results.

- It is suggested to incorporate the strengths and limitations of the research into the discussion.

Additional comments

No comments

---

## Round 0.2 · accepted · Accept

In congratulate the authors on a well improved manuscript. All reviewer comments have been adequately addressed.

Reviewer 1 ·

Basic reporting

ok

Experimental design

ok

Validity of the findings

ok

Additional comments

no

·

Basic reporting

I appreciate you taking the time to apply the suggested changes to the manuscript. These modifications make the work more robust and help the reader better understand the study.

Experimental design

No comment

Validity of the findings

No Comment

Additional comments

No comment.